# Improving Speech Recognition Performance in Noisy Environments by Enhancing Lip Reading Accuracy

**DOI:** 10.3390/s23042053

**Published:** 2023-02-11

**Authors:** Dengshi Li, Yu Gao, Chenyi Zhu, Qianrui Wang, Ruoxi Wang

**Affiliations:** School of Artificial Intelligence, Jianghan University, Wuhan 430056, China

**Keywords:** audiovisual speech recognition, noisy environment, lip reading, cross-modal fusion

## Abstract

The current accuracy of speech recognition can reach over 97% on different datasets, but in noisy environments, it is greatly reduced. Improving speech recognition performance in noisy environments is a challenging task. Due to the fact that visual information is not affected by noise, researchers often use lip information to help to improve speech recognition performance. This is where the performance of lip recognition and the effect of cross-modal fusion are particularly important. In this paper, we try to improve the accuracy of speech recognition in noisy environments by improving the lip reading performance and the cross-modal fusion effect. First, due to the same lip possibly containing multiple meanings, we constructed a one-to-many mapping relationship model between lips and speech allowing for the lip reading model to consider which articulations are represented from the input lip movements. Audio representations are also preserved by modeling the inter-relationships between paired audiovisual representations. At the inference stage, the preserved audio representations could be extracted from memory by the learned inter-relationships using only video input. Second, a joint cross-fusion model using the attention mechanism could effectively exploit complementary intermodal relationships, and the model calculates cross-attention weights on the basis of the correlations between joint feature representations and individual modalities. Lastly, our proposed model achieved a 4.0% reduction in WER in a −15 dB SNR environment compared to the baseline method, and a 10.1% reduction in WER compared to speech recognition. The experimental results show that our method could achieve a significant improvement over speech recognition models in different noise environments.

## 1. Introduction

Language is the most natural, effective, and intuitive way for humans to express themselves to each other; through speech, humans can obtain and understand more information. With the development of deep learning in recent years, great success has been achieved in the field of speech recognition [1,2]; currently, speech recognition is rather accurate with a word error rate (WER) of less than 3% in quiet environments according to different datasets. However, in noisy environments, the performance of speech recognition is drastically degraded by noise, and it is difficult to achieve significant improvement by using noise reduction and speech separation methods [3,4].

Audiovisual speech recognition (AVSR) is a task that leverages both the audio input of human voices and the aligned visual input of lip motions. In recent years, it has been one of the most successful application fields that involve multiple modalities. It uses lip motion information to assist in speech recognition and improve speech recognition performance in noisy environments. Therefore, the performance of lip reading plays a significant role. However, most of the lip reading modules of existing audiovisual speech recognition models use only a single piece of information, such as lip pictures, to achieve this, ignoring the fact that the same lip may contain multiple meanings and lip morphology and phonemes are in a one-to-many relationship. For example, the “mat” and “bat” lip shapes are the same, but pronounced differently. If there is no tongue position information, it is difficult to distinguish them by using visual features alone, which causes information recognition errors. In 2022, Qiya Song et al. [5] proposed adding optical flow to assist in lip reading, which improved lip reading and audiovisual speech recognition performance. Therefore, it is possible to improve lip reading and thus the accuracy of audiovisual speech recognition without introducing a new input and using only the audio part in the audiovisual combination.

Multimodal fusion is also an important part of the audiovisual speech recognition field. For the current case with inputs of both visual and audio modalities considered, it reduces to a two-modal fusion problem. Most previous studies have used simple concatenation [6,7,8], and a few others have used attention mechanisms to achieve fusion [5]. However, intra- and inter-modal relationships are ignored in fusion, such as multimodal emotion recognition, video retrieval, and video segmentation. Many different methods for multimodal fusion have been proposed; notably, Praveen et al. [9] proposed a cross-joint attention fusion model to achieve emotion recognition in 2022. Experiments verified the effectiveness of the fusion module and improved the performance of the emotion recognition model.

Although the performance of lip reading is particularly important, the performance of existing lip reading models for audiovisual speech recognition is poor due to the fact that they only focus on the mapping relationship from lips to words, even though lips to words are not one-to-one mapping. Multimodal fusion is also a still-unsolved problem.

This study aims to improve the accuracy of speech recognition in noisy environments with the assistance of lip reading, which could effectively reduce the speech recognition error rate, since lip information is not affected by noise and thus is most suitable for recognizing speeches in acoustic noisy environments.

The main contributions of this work can be summarized as follows: (1) a lip-to-audio one-to-many mapping model has been designed for extracting accurate visual features as the encoding part of the lip recognition model; (2) a joint cross-fusion model has been designed for the fusion of cross-modal features, with multiple attention weights computed on the basis of the attention mechanism that can effectively obtain the correlations between different modalities and within the membrane state; (3) with four different types of background noise added, our simulational experiments show that our method can reduce the WER of speech recognition compared to the baseline method.

## 2. Related Work

### 2.1. Lip Reading

Lip reading is used to infer what the speaker is saying through analyses of a series of information on lip movement. Assael et al. [10] developed the first end-to-end sentence-level visual speech recognition system, which achieved the best performance on the GRID dataset, extracting video features through a 3D spatiotemporal convolutional network and completing sequence modeling using a gated recurrent unit. In 2017, Chuang et al. [11] also developed an end-to-end sentence-level visual speech recognition system on the basis of an attention-based sequence-to-sequence model in wilderness scenes. In 2020, Martinez et al. [12] improved temporal encoding by proposing a multiscale temporal convolutional network (MS-TCN), and boosted word-level lip reading performance. In 2022, Koumparoulis et al. [13] presented a resource-efficient end-to-end structure and introduced efficient nets to lip reading. The best performance to date was obtained on the LRW dataset without additional training data, with an accuracy of 89.52%; with the help of additional training data, the current record high is 94.1% [14]. Without using additional training data and language models, the best model so far has been the one (based on the LRS2 dataset) proposed by Pingchuan Ma in 2022 [15] with a WER of only 32.9%. Some studies have focused on bringing audio modal information into visual modality. They successfully complemented the insufficient speech information of lip videos with rich audio information. For example, the authors in [16] proposed a visual–audio memory algorithm, which could recall audio features from the input video and achieve better performance on the LRW dataset, with an accuracy of 88.5%. The algorithm can help to solve the problem that lip syntheses may contain multiple meanings.

### 2.2. Audiovisual Speech Recognition

In 2018, Afouras [6] developed a transformer-based sequence-to-sequence audiovisual speech recognition model using precomputed visual features and audio Log-Mel filter features as inputs, achieving a state-of-the-art performance at that time. The field of audiovisual speech recognition has been rapidly developing since then. In 2020, George Sterpu et al. [17,18] proposed an AV-Align network with an attention mechanism to make the network focus more on audio modality in audiovisual speech recognition, gaining significant improvements. Later, further developments were made in application scenarios to study audiovisual speech recognition for multiple and sometimes overlapping speeches. For example, Otavio Braga et al. [19] considered the case of multi-speakers and used an attention mechanism to calculate the similarity, find the speaker’s facial changes corresponding to the audio and realize audiovisual speech recognition with multiple faces appearing on the screen. Jianwei Yu et al. [20] also investigated the use of audiovisual technologies for overlapped speech recognition. In 2021, Pingchuan Ma et al. [7] made end-to-end learning on LRS2 a possibility by using a conformer acoustic model and a hybrid CTC/attention decoder, achieving even better recognition results. Xichen Pan et al. [21] used two single-modal self-supervised modules of wav2vec and MoCo for cross-modal self-supervised audiovisual speech recognition, which achieved the best performance so far on the LRS2 [11] dataset without using language models, with a WER of only 2.7% for speech recognition and 2.6% for audiovisual speech recognition.

## 3. Architecture and Methods

The overall architecture of our audiovisual model, as shown in Figure 1, has the following modules.

### 3.1. Automatic Speech Recognition

The model boxed with black dashed lines in Figure 1 is an automatic speech recognition model, it used the transformer network with noisy speeches as inputs and the corresponding texts as outputs.

#### 3.1.1. Audio Frontend

The wav2vec 2.0 [22], pre-trained on Libri-Light [23], was adopted to transfer both the 1D convolutional layers and the stacked transformer encoder layers into our audio frontend, following the usual ASR procedures. In this study, raw audio waves of 16 kHz with multiple audio data frames (each containing 20 ms of audio data) were used as inputs for the audio frontend. More details can be found in Table 1.

#### 3.1.2. Audio Backend

To unify the feature dimensions of the two modalities, we used 1D convolution to transform the feature dimensions to 512, and the resulting feature is denoted as $f_{a}$. The transformer network was used as the main model for speech recognition to achieve speech-to-text conversion.

### 3.2. Visual Speech Recognition

As one of the most important modules of audiovisual speech recognition, visual speech recognition plays a key role in its performance. We have chosen to add clean audio to help training lip reading to improve the accuracy of visual speech recognition. At the inference stage, where only lip video input is available, we can extract the saved audio features from the memory by examining the learned inter-relationships using the input visual features.

#### 3.2.1. Visual Frontend

Visual frontend serves as a component to capture the lip motion and reflect the lip position differences in its output representations. Here, we have followed the same procedures as Xichen Pan [21]. We have truncated the first convolutional layer in MoCo v2, which was pre-trained on ImageNet, and replaced it with a 3D convolutional layer. Intentionally, the output of the 3D convolutional layer is the same as the input of the first ResBlock in MoCo v2, so that deeper features can be extracted. On the other hand, we have also adopted the common practice of converting RGB input images to grayscale, as it prevents the model from learning chromatic information. The visual feature dimensions are shown in Table 2.

#### 3.2.2. Visual Backend

Again, we have used 1D convolution to transform the feature dimensions to 512, and the resulting feature is denoted as $f_{v}$. Since one mouth shape may correspond to multiple articulations, a one-to-many mapping relationship between lip movements and audio must be determined beforehand. In this study, we have adopted Kim’s [16] multi-head K-V memory network model to describe the one-to-many mapping relationship.

The multi-head K-V memory network consists of two parts, which are multi-head key memory $K=K_{v1},...,K_{vh}$, where $K\in R^{N\times \frac{D}{h}}$, and value memory, $V\in R^{N\times D}$, where *N* is the number of memory slots and D is the model dimension. The obtained visual features, $f_{v}$, are fed into the key memory network which is used to store the visual features. The obtained audio features, $f_{a}$, are fed into the value memory network, which is used to store the audio features. Similar to the multi-head attention [24], we have set the Key Memory as the network of multiple heads to preserve multiple possible lip features. Since the key memory and value memory are trained to save and to read the features of paired audiovisual data, it is possible to obtain the saved audio features by using visual inputs only. The corresponding addressing scores are then obtained by a similarity-based addressing [25,26] calculation:(1)$$\begin{array}{cc} A_{vh}^{i,j} & =\frac{exp(\alpha \cdot d\left(K_{vh}^{i},W_{qh}^{T}f_{v}^{j}\right))}{\sum_{n=1}^{N}exp\left(\alpha \cdot d\left(K_{vh}^{n},W_{qh}^{T}f_{v}^{j}\right)\right)} \end{array}$$
where d(·) is a cosine similarity metric, $W_{qh}\in R^{N\times \frac{D}{h}}$ represents the projection weight for h-th head, and α is a scaling factor. Extraction of *h* possible different audio features from the value memory is executed by the obtained addressing score.
(2)$$\begin{array}{cc} a_{h}^{j} & =\sum_{i=1}^{N}A_{vh}^{i,j}\cdot M_{a}^{i}\\ a^{j} & =Concat(a_{1}^{j},\ldots ,a_{h}^{j})\\ {f_{a}^{j}}^{\prime } & =W_{o}^{T}a^{j}\\ \tilde{f_{v}} & =f_{v}+{f_{a}}^{\prime } \end{array}$$
where $a_{h}^{j}$ represents extracted audio features from the value memory using the *h*-th head key memory and $W_{o}\in R^{Dh\times D}$ is the embedding weight that aggregates the *h* different extracted audio features. The obtained features, ${f_{a}^{j}}^{\prime }$, are then summed with the input visual features, $f_{v}$, and layer normalized to obtain the final visual features, $\tilde{f_{v}}$.

### 3.3. Joint Cross-Modal Fusion Module

Visual modality and audio modality belong to two different types of modal features. How to use visual modality to assist audio modality features to improve the accuracy of speech recognition in noisy environments is a key issue. We need to learn information from visual modality features related to audio modality features to build a fusion module to achieve focus on inter- and intra-modal interactions between different modalities.

Let $\tilde{f_{v}}\in R^{B\times D}$ and $\tilde{f_{a}}\in R^{B\times D}$ represent two sets of deep feature vectors extracted for visual and audio modalities, where B denotes the size of the batchsize. Here, both $\tilde{f_{v}}$ and $\tilde{f_{a}}$ are one-dimensional. Since the length of each video sequence is inconsistent, we fix the feature dimension to 512 and compute it *l* times, where *l* represents the feature length. A joint representation of audiovisual features, $\tilde{f_{av}}$, is obtained by concatenating the visual and audio feature vectors $\tilde{f_{av}}\in R^{B\times 2D}$, as shown in Figure 1:(3)$$\begin{array}{cc} \tilde{f_{av}} & =[\tilde{f_{v}};\tilde{f_{a}}] \end{array}$$

The joint correlation matrix across the visual features, $\tilde{f_{v}}$, and the combined audiovisual features, $\tilde{f_{av}}$ is given by:(4)$$\begin{array}{cc} H_{v} & =tanh(\frac{{\tilde{f_{v}}}^{T}W_{jv}\tilde{f_{av}}}{\sqrt{D}}) \end{array}$$
where $W_{jv}\in R^{B\times B}$ represents the learnable weight matrix across visual and joint audiovisual features. Similarly, the joint correlation matrix for audio features is given by:(5)$$\begin{array}{cc} H_{a} & =tanh(\frac{{\tilde{f_{a}}}^{T}W_{ja}\tilde{f_{av}}}{\sqrt{D}}) \end{array}$$
where $W_{ja}\in R^{B\times B}$ represents the learnable weight matrix across visual and joint audiovisual features.

The joint correlation matrix of the computational visual and audio modalities provides semantic correlation within the same modality, and higher correlation coefficients indicated that the corresponding data samples are strongly correlated with other modalities. Therefore, the method can effectively exploit the inter-relationship between visual and audio modalities, and thus improve the performance of the system. After computing the joint correlation matrices, the attention weights of A and V modalities were estimated. We rely on different learnable weight matrices corresponding to features of the individual modalities to compute the attention weights of the modalities.

For the visual modality, we have used the obtained correlation weight matrix, $H_{v}$, with the learnable weight matrix to calculate the corresponding attention weights $Z_{v}$:(6)$$\begin{array}{cc} Z_{v} & =relu(W_{v}\tilde{f_{v}}+W_{hv}H_{v}^{T}) \end{array}$$
where $W_{hv}\in R^{2D\times B}$, $W_{v}\in R^{B\times B}$, and $Z_{v}$ represent the attention weight of the visual modality. Similarly, the attention maps of the audio modality are obtained as:(7)$$\begin{array}{cc} Z_{a} & =relu(W_{a}\tilde{f_{a}}+W_{ha}H_{a}^{T}) \end{array}$$
where $W_{ha}\in R^{2D\times B}$, $W_{a}\in R^{B\times B}$, and $Z_{a}$ represent the attention maps of the audio modality.

Finally, the attention maps are used to compute the attended features of audio and visual modalities. These features are obtained as:(8)$$\begin{array}{cc} \bar{f_{v}} & =W_{zv}Z_{v}+\tilde{f_{v}}\\ \bar{f_{a}} & =W_{za}Z_{a}+\tilde{f_{a}} \end{array}$$
where $W_{zv}\in R^{B\times B}$ and $W_{za}\in R^{B\times B}$ denote the learnable weight matrices, respectively. The obtained $\bar{f_{v}}$ and $\bar{f_{a}}$ are concatenate together, which is given by:(9)$$\begin{array}{cc} \bar{f_{av}} & =[\bar{f_{v}};\bar{f_{a}}] \end{array}$$

Finally, we have obtained the fused features through the designed cross-fusion module, $\bar{f_{av}}$, and at this time, the feature dimensions were not consistent with the model dimensions that are required to feed into the network. Therefore, we added a linear layer for converting the dimensions, converting the 1024-dimensional feature dimensions to 512-dimensional, and then performing encoding and decoding operations through the underlying transformer network to achieve speech recognition. The fusion feature dimensions are shown in Table 3.

### 3.4. Decoders

Following the settings of Petridis et al. [27], there are two decoders trianed simultaneously.

The first was the transformer seq2seq decoder, which used a 6-layer transformer decoder. We used ground-truth characters as inputs during training and performed teacher forcing at the character level.

The second was arguably a decoder, because it produced a character probability for each time step and was dependent on the CTC loss [28] in training. Four additional one-dimensional convolutional layers with ReLU activation were used above the output of the last transformer encoder layer. We also included layer specification between each layer.

### 3.5. Loss Function

In this work, we have used four loss functions for training. The first was used for the training of the lip reading model. In order to preserve a representative audio representation in the value memory network, reconstruction-based and contrast-based learning were used to train the value memory. Reconstruction-based contrast loss was also used in order to ensure that the correct form of audio information is preserved in the value memory. It is defined as a reconstruction loss on the basis of the cosine similarity:(10)$$\begin{array}{cc} L_{rec} & =||1-d\left(\hat{f_{a}},f_{a}\right){\left|\right|}_{1}\\ \hat{f_{a}} & =\sum_{i=1}^{N}A_{a}^{i,j}\cdot V^{i} \end{array}$$
where $\hat{f_{a}}$ represents the reconstructed audio features from the value memory by using the addressing score. Note that $A_{a}^{i,j}$ is obtained similar to Equation (Equation 1) by substituting key memory and visual features with value memory and audio features. Figure 2 shows the training process of the value memory.

The contrast loss, used to preserve different representative audio features, is defined as follows:(11)$$\begin{array}{cc} L_{cont} & =\sum_{i\neq j}||1-d\left(V^{i},V^{j}\right){\left|\right|}_{1} \end{array}$$

Contrast loss leads to different memory slots that have less similar audio characteristics, which leads the value memory possibly containing discriminative audio representations.

Then, for the training of the three models, we used a so-called hybrid CTC/attention loss [29]. Let $x=[x_{1},\ldots ,x_{T}]$ be the input frame sequence and $y=[y_{1},\ldots ,y_{L}]$ be the targets, where *T* and *L* denote the input and target lengths, respectively.

Assuming that the outputs of each time series are independent of each other, the posterior probability of the path is the accumulation of the probabilities of each time series.
(12)$$\begin{array}{c} P_{CTC}\left(y|x\right)\approx \prod_{t=1}^{T}P\left(y_{t}|x\right) \end{array}$$

On the other hand, an auto-regressive decoder removes this assumption by directly estimating the posterior on the basis of the chain rule. It uses cross entropy loss.
(13)$$\begin{array}{cc} P_{CE}\left(y|x\right) & =\prod_{t=1}^{T}P\left(y_{l}|y_{<l},x\right) \end{array}$$

The overall objective function is computed as follows:(14)$$\begin{array}{cc} L & =L_{rec}+L_{cont}+\lambda logP_{CTC}\left(y|x\right)+\left(1-\lambda \right)logP_{CE}\left(y|x\right) \end{array}$$
where λ controls the relative weight between CTC loss and seq2seq loss in the hybrid CTC/attention mechanisms.

### 3.6. Training Pipeline

Our overall training pipeline is shown in Figure 3.

The goal of this study is to improve the performance of speech recognition in noisy environments, where the input is noisy speech and the output is a speech model of text, as shown in the black dashed box in Figure 3. The frontend of the audio is trained using the already trained wav2vec 2.0 model, and then the backend of the audio and the decoder part is trained by the audio-only (AO) setup.

Next is the use of lips to help the audio for speech recognition, and then the lip reading model needs to be trained. The frontend of visual needs to be pre-trained with the LRW dataset and then the backend and decoder part of visual is trained by the video-only (VO) setup.

The final AVSR model, the frontend and backend of both visual and audio are trained, and the trained features are extracted from the AO and VO training models. Only the parameters of the fusion module and decoder need to be learned in the final stage.

### 3.7. Decoding

Decoding is performed using joint CTC/attention [24] one-pass decoding with beam search.
(15)$$\begin{array}{cc} \hat{y} & =\underset{y\in \hat{y}}{\arg\max}\alpha logP_{CTC}\left(y|x\right)+\left(1-\alpha \right)logP_{CE}\left(y|x\right) \end{array}$$
where $\hat{y}$ denotes the predicted set of target symbols, while α is the relative weight that is tuned on the validation set.

## 4. Experiments

### 4.1. Dataset and Evaluation Metric

We used a large-scale public AVSR dataset, namely Lip Reading Sentences (LRS2), as our main training and testing dataset. During training, we also used Lip Reading in the Wild (LRW) as a word-level video classification task to pre-train our visual frontend.

LRW [30] is a word-level dataset which includes up to 1000 discourses, each consisting of 500 different words spoken by hundreds of different speakers. It contains 157 h of aligned audio and video, for a total of 489K BBC video clips. All videos are 29 frames (1.16 s) in length, with the words appearing in the middle of the video. Word durations are given in the metadata, from which the start and end frames are determined. In our experiment, we only used the visual modality from this dataset to train our visual frontend.

LRS2 [11] is a sentence-level dataset, it contains thousands of spoken sentences from BBC television, including 224 h of aligned audio and video with a total of 144K BBC video clips. The training set, validation set, and test set are divided according to the broadcast date. The utterances in the pre-training set correspond to partial sentences and multiple sentences, while the training set contains only single complete sentences or phrases. There are some overlaps between the pre-training and training sets. The dataset is very challenging due to the large variation in head poses, lighting conditions, genres, and the number of speakers.

With the word error rate (WER), a common evaluation criteria for AVSR, we can evaluate the effectiveness of our method. The WER can be expressed as WER=(S+D+I)/Num, where Num is the number of words in the reference and *S*, *D*, and *I* are the numbers of substitution, deletion, and insertion operations to re-edit the hypothesis sentence to be exactly the same as the reference one.

### 4.2. Experimental Settings

We used character-level predictions with an output size of 40, including 26 characters, 10-digit numbers, apostrophes, and special tags for [spaces], [blanks], and [EOS/SOS] in the alphabet. Since the transcription of the dataset did not contain other punctuation marks, we did not include them in the vocabulary.

We implemented it on the basis of the pytorch framework [31]. The network is trained using the Adam optimizer [32], with $\beta_{1}=0.9$, $\beta_{2}=0.999$, and $\epsilon =10^{-7}$ and an initial learning rate of $10^{-4}$. We used a threshold with weight set to 0.001, learning rate warm up, and reduce on plateau scheduler. The relative weight in CTC loss and CE loss λ was set to 0.2. When decoding, we set α to 0.1.

During preprocessing of the data, we detected and tracked 68 facial landmarks for each video using dlib [33]. To eliminate the differences associated with face rotation and scale, we have used interpolation with a window width of 12 frames and frame smoothing to handle frames that could not be detected by dlib. Then, a 120 × 120 bounding box was used to crop the oral roi. The cropped frames were further converted to grayscale and normalized on the basis of the overall mean and variance of the training set. The samples of the pretraining set were clipped by randomly selecting 1/3 words of the entire discourse in a continuous range to match the fragment length in the training set. Overly long samples were further truncated at 160 frames to reduce the memory footprint. Each original audio waveform was normalized to zero mean and unit variance [34].

For the processing part of the data enhancement, a random crop size of 112 × 112 and a horizontal flip of probability 0.5 were performed consistently across all frames of a given image sequence [7]. For each audio waveform, additional noise was performed in the time domain after Afouras et al. [6], by adding Babble noise to the audio stream with a 5 dB SNR and a probability of $p_{n}=0.25$. The Babble noise was synthesized by mixing 20 different audio samples from LRS2.

### 4.3. Experimental Results

#### 4.3.1. Comparison Experiments

Our first experiment compared the respective WERs of the three models, VO, AO, and AV, for a comparison of the different methods in the presence of clean speech, and all the experimental results are shown in Table 4.

For visual-only methods, we used the LRS2 dataset as our primary training and testing dataset. First we compared models in the field of lip reading, such as LIRS [35] and MVM [16], and we can see that the WER of our method is only 40.1%, which is a 4.4% improvement compared to our reference MVM, and a more than 25% improvement compared to LIBS proposed in 2020. Second, we compared the lip reading models in the field of audiovisual speech recognition in recent years, also exhibiting a 2.3% improvement compared to the best E2E conformer [7] model.

For audio-only methods, we used the same LU-SSL transformer method proposed by Xichen Pan et al. [21] in 2022, so the WER is consistent with that method, with an error rate of only 2.7%, which is currently the best achieved on the LRS2 dataset.

For audio-visual methods, we compared the four methods before 2021, and found that all such results were improved by 1.4% with respect to the best E2E conformer [7] method. Then, we can conclude that our method has the same performance as the baseline method.

#### 4.3.2. Noise Environment Comparison Experiments

In the second experiment, we tested the WERs of the AV and AO models in different SNR environments, as shown in Table 5.

This experiment uses babble noises, which include not only noisy speech but also background noises of multiple speakers, and are made available from our LRS2 dataset.

Firstly, we can see that for the speech recognition AO model, as the signal-to-noise ratio decreases, the WER increasingly rises and reaches 32.5% when the SNR is 0 dB, which also shows that in the face of a noisy environment, the performance of speech recognition decreases substantially, and the word error rate rises by 29.8% compared to a clean environment. Secondly, for the audiovisual speech recognition model, it can be seen that there is a significant improvement, with an SNR of 0 dB, and the WER is reduced to 22.4%. Furthermore, compared with the baseline model, our method has different degrees of improvement in different signal-to-noise ratios, and the higher the noise, the more obvious the improvement in speech recognition performance. The experiments show that our method can effectively improve the performance of speech recognition in noisy environments.

In addition to the improvement over the comparison method in babble noise environments, we have further investigated the performance of the model in human noise environments. Human noise is the sound made by multiple speakers speaking at the same time, and the noise in which the meaning of the utterance can be clearly heard. We synthesized human noise by randomly cropping many one-second signals from different audio samples in the LRS2 dataset. Human noise is extremely challenging because the noise itself contains a number of words, and the model can not easily distinguish which audio signals are to be recognized, as shown in Table 6.

Here, we have chosen five different human noise levels, with an SNR of −5 dB to 15 dB, and it can be seen that our models have all improved significantly compared to speech recognition, and the WER has been reduced by more than 10% below the SNR of 0 dB, indicating that our proposed method can effectively reduce the word error rate of speech recognition in a strong noise environment. Furthermore, compared with the baseline method, our methods are all improved, and it can be seen from the table that the higher the noise, the more significant the improvement of our model.

In the third experiment, we aimed to test the extent to which speech recognition performance is affected under different kinds of noise and to verify the auxiliary effect of visual information on the performance. To standardize the same signal-to-noise ratio, we chose babble noise, human noise, industrial noise, and white noise for the test, as shown in Figure 4.

Firstly, it can be seen from Figure 4 that the combined audiovisual model proposed by us outperforms the speech recognition model in different kinds of noise environments and different SNR environments, indicating that our method can effectively improve the accuracy of speech recognition in noisy environments. Secondly, the WER of speech recognition of human noise for our method is the highest for all four different SNR environments, indicating that our method is the most suitable for the case of background noise containing specific information. In addition, the WER increases significantly when the signal-to-noise level drops below 0 dB, which is because the model may not be able to distinguish the two overlapping spoken words at low signal-to-noise levels. Thirdly, we found that the speech recognition error rate in industrial noise background is higher than that in babble noise background when the SNR level is above 5 dB, while the speech recognition error rate with an industrial noise background is lower than that with a babble noise background when the SNR level is below 0 dB. This indicates that the disordered nagging human voice affects the speaker’s speech recognition more in the environment with strong noises. If the babble noise volume is not high, industrial noise affects speech recognition more. Fourth, when the SNR is below −10 dB or less, the performance of our proposed model shows little improvement in the environment of human noise, indicating that human noise has the greatest effect on speech. Finally, when the noise is very large or small, the improvement is not significant compared to the speech recognition model in the context of industrial noise.

In general, our experiments were tested, giving the following results: (1) the effectiveness of the proposed lip reading recognition model has been verified, and the WER is reduced by 2.3% compared with the latest model; (2) the performance of audiovisual speech recognition in a clean environment has been verified, and the performance is on par with the baseline model; (3) we have tested the WER in a babble noise environment and a human noise environment with an SNR of 0 dB, and found that the proposed model’s WER is 2.1% and 1.7% lower than that of the baseline model, respectively; and (4) we have compared our model and speech recognition model under four different environmental noise types (babble noise, human noise, industrial noise, and white noise), showing that the method proposed in this paper could achieve a significant improvement in different noise scenarios, and this performance improvement is significant in the environments of babble noise and white noise.

## 5. Conclusions

In this paper, we have proposed the multi-head K-V memory model for lip reading and the joint cross-modal fusion model for fusion. The multi-head K-V memory model includes a one-to-many mapping relationship from lips to audio, and can extract well-preserved audio representations from memory through mutual relationships with visual-only input, compensate for the lack of visual features with audio information, and validate the effectiveness of the model experimentally on large-scale sentence-level datasets. The joint cross-modal fusion model can learn the inter- and intra-modal inter-relationships, which effectively improves the cross-modal fusion effect and improves the performance of audiovisual combinations in the final experimental results by decreasing the WER in noisy environments. Moreover, it is found that the speech recognition performance of our methods is least affected by babble noise.

## Figures and Tables

**Figure 1 sensors-23-02053-f001:**
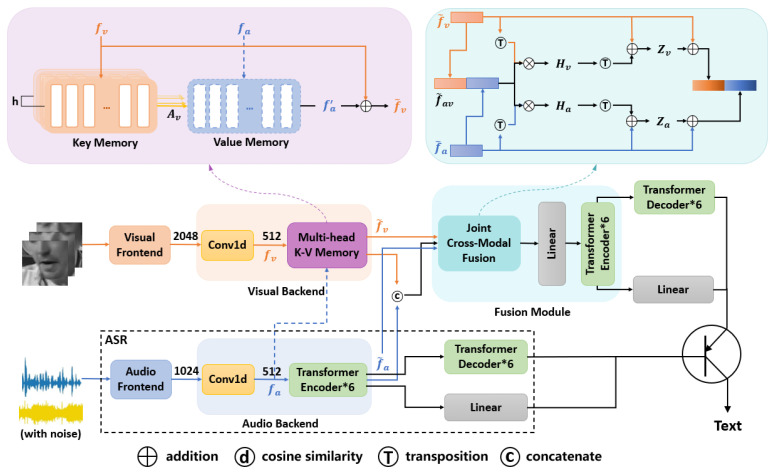
Overall architecture of our AVSR model. The orange lines in the figure all represent visual input, the blue lines all represent audio input, and some of the blue dashed lines represent use during training only. The circle boxed out represents the switch symbol, representing the addition of lip-synthesis to a noisy speech recognition model to improve performance.

**Figure 2 sensors-23-02053-f002:**
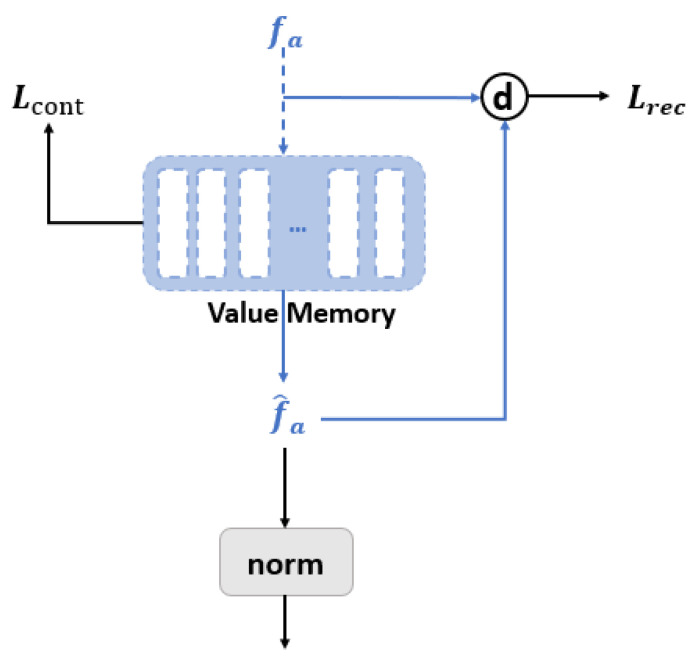
Learning to save audio representations into value memory with reconstruction and contrastive losses.

**Figure 3 sensors-23-02053-f003:**
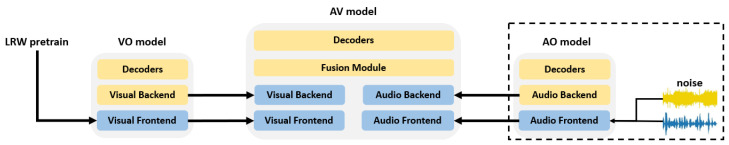
Training pipeline of the model. Blue blocks represent new parameters that are randomly initialized, while yellow blocks represent parameters that are inherited from last training stage.

**Figure 4 sensors-23-02053-f004:**
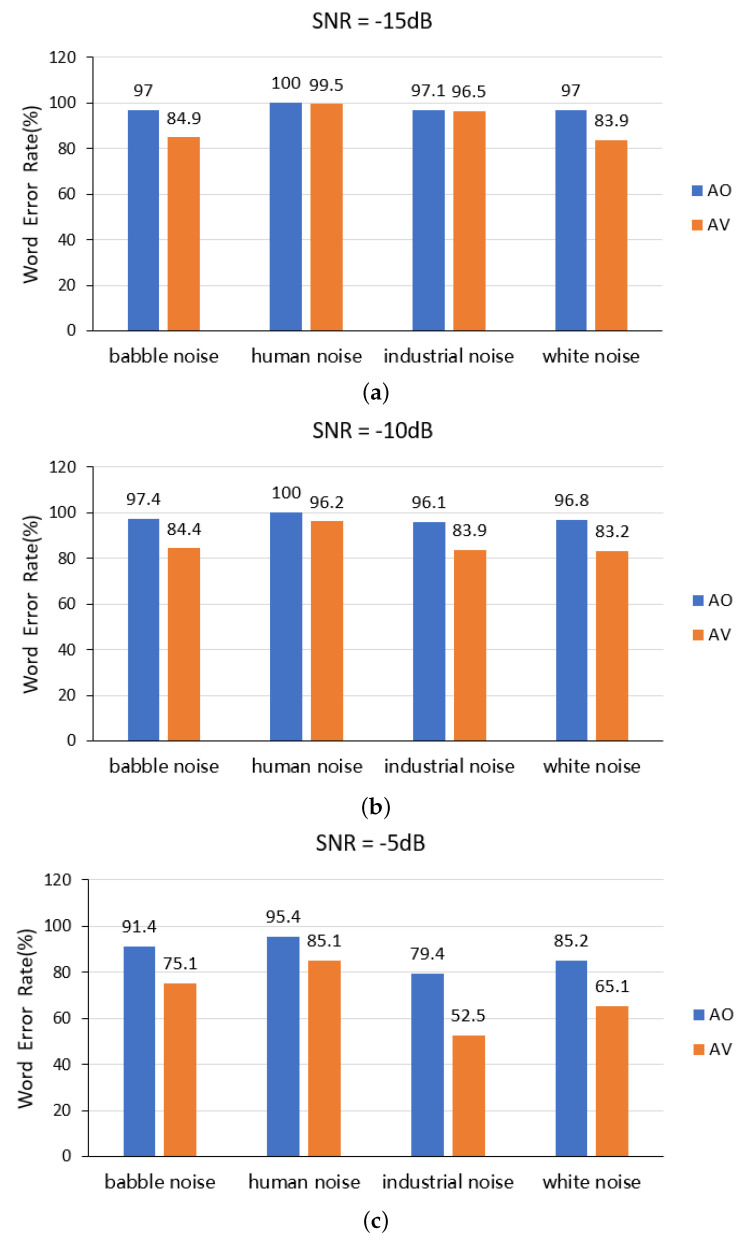
WER comparison of speech recognition and audio-visual speech recognition models under different signal-to-noise ratios and noise types.

**Table 1 sensors-23-02053-t001:** This feature dimension of audio stream.

Stage	Modules	Audio Waveform $\left(T_{a}\times 1\right)$
Audio Frontend	wav2vec 2.0	$T_{v}\times 1024$
Audio Backend	1D convolution	$\frac{T_{v}}{2}\times 512$
	Transformer Encoder	$\frac{T_{v}}{2}\times 512$

**Table 2 sensors-23-02053-t002:** This feature dimension of visual stream.

Stage	Modules	Image Sequence ($T_{v}\times 112^{2}\times 1$)
Visual Frontend	3D convolution	$T_{v}\times 28^{2}\times 64$
	MoCo v2	$T_{v}\times 2048$
Visual Backend	1D convolution	$T_{v}\times 512$
	Multi-head K-V Memory	$T_{v}\times 512$

**Table 3 sensors-23-02053-t003:** This feature dimension of fusion feature.

Stage	Modules	Fusion Features
Fusion Module	Joint Cross-Modal Fusion	$\frac{T_{v}}{2}\times 1024$
	Linear	$\frac{T_{v}}{2}\times 512$
	Transformer Encoder	$\frac{T_{v}}{2}\times 512$

**Table 4 sensors-23-02053-t004:** Audio-only, visual-only and audio-visual results of word error rate (WER) tested on LRS2. Models with an * denote that results are using an external language model, which indicates an advantage over our model during evaluation. The arrow (↓) indicates that the lower the WER, the better the model performance.

Methods	WER (%)
Visual-only (VO) (↓)	
LIBS [35]	65.3
TM-CTC * [6]	54.7
Conv-seq2seq [36]	51.7
TM-seq2seq * [6]	50.0
LF-MMI TDNN * [20]	48.9
MVM [16]	44.5
LU-SSL Transformer [21]	43.2
E2E Conformer * [7]	42.4
Ours	**40.1**
Audio-only (AO) (↓)	
TM-CTC * [6]	10.1
TM-seq2seq * [6]	9.7
CTC/attention * [27]	8.2
LF-MMI TDNN * [20]	6.7
E2E Conformer * [7]	4.3
LU-SSL Transformer [21]	2.7
Ours	**2.7**
Audio-Visual(AV)(↓)	
TM-seq2seq * [6]	8.5
TM-CTC * [6]	8.2
LF-MMI TDNN * [20]	5.9
E2E Conformer * [7]	4.2
LU-SSL Transformer [21]	2.8
Ours	**2.8**

**Table 5 sensors-23-02053-t005:** WER under different SNR levels. The noises are synthesized babble noises.

Modality	Model	15 dB	−10 dB	−5 dB	0 dB	5 dB	Clean
AO	Afouras et al. [6]	-	-	-	58.5%	-	10.5%
	Ours	97%	97.4%	91.4%	**32.5%**	7.2%	**2.7%**
AV	Afouras et al. [6]	-	-	-	33.5%	-	9.4%
	Xichen Pan et al. [21]	88.9%	88.2%	77.1%	24.5%	6.3%	2.8%
	Ours	**84.9%**	**84.4%**	**75.1%**	**22.4%**	**5.9%**	**2.8%**

**Table 6 sensors-23-02053-t006:** WER under different SNR levels. The noises are synthesized human noises.

Modality	Model	−5 dB	0 dB	5 dB	10 dB	15 dB
AO	Ours	**95.4%**	**69.3%**	**26.2%**	**7.8%**	**3.9%**
AV	Xichen Pan et al. [21]	87.4%	58.9%	20.9%	6.6%	3.6%
	Ours	**85.1%**	**57.2%**	**20.7%**	**6.0%**	**3.5%**

## Data Availability

Not applicable.

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
