# Peer review of "Improving Speech Recognition Performance in Noisy Environments by Enhancing Lip Reading Accuracy"

_sensors, 2023, doi:10.3390/s23042053_

Round 1

Reviewer 1 Report

2. Summary. In 5-7 sentences, describe the key ideas, experimental or theoretical results, and their significance.

In this paper, the author improves the performance of speech recognition in noisy environments by building a lip-to-speech one-to-many mapping model and a joint fusion module. Since lips and text are not in a one-to-one mapping relationship, using audio to select the correct lips and saving the corresponding audio features improves lip-reading performance. Focus on intra-modal and inter-modal similarity to learn the interrelationship between audio and visual modalities to achieve improved speech recognition accuracy in different noise environments. This work is of interest and the experimental results are improved compared to the baseline method.

3. Strengths. Consider the significance of key ideas, experimental or theoretical validation, writing quality, data contribution. Explain clearly why these aspects of the paper are valuable. Short bullet lists do NOT suffice.

The key idea of constructing one-to-many mapping relationships with audio in audio-visual speech recognition is interesting. This work is of great interest in the field of speech recognition. The experimental results confirm the effectiveness of the proposed method. In terms of the paper writing, the overall logic is relatively clear and the lines are more fluent. Overall, the paper flows well and is reasonably prepared.

4. Weaknesses. Consider the significance of key ideas, experimental or theoretical validation, writing quality, data contribution. Clearly explain why these are weak aspects of the paper, e.g. why a specific prior work has already demonstrated the key contributions, or why the experiments are insufficient to validate the claims, etc. Short bullet lists do NOT suffice.

1) In Section 3, the author used D in the feature dimension representation, but did not explain the meaning of it, and suggested to add it.

2) In the eighth formula, the author has a writing error and the subscripts of the weight matrix are misrepresented, which is suggested to be revised.

3) In Section 3.6, the author has written Figure 3 as Figure 2 and have not added a citation, which is recommended to be revised.

Author Response

Thank you very much for the advice you gave me, mainly the errors in expression and writing, which I have revised in the paper.

Reviewer 2 Report

This study is well-motivated: improving speech recognition in noise by using information from the lips is potentially very useful, it's something that human listeners do and there is a body of previous work which the authors review adequately

The English is mostly OK but I am attaching a copy of the paper in which I have highlighted phrases which are badly formed or unclear. On the same copy a comment ?? indicates something I don't understand.

The section on acoustic-phonetics, lines 36-46, is confused and should be re-written with the aid of a phonetics textbook. You should use IPA symbols not "th" etc.

At different points you talk about using information from both lips and the fact that lip-based information cannot help to distinguish between all phonemes. The way you've expressed this makes it easy to get confused between the two

Your experiments use speech plus added noise. This makes for neat experiments but is known to be unrealistic: sound sources don't combine in such a simple way.  That's particularly the case in AVSR because if people are speaking in noise they will change their vocal effort . the Lombard effect.

In figure 1 there appears to be a transistor at the bottom right :-)

Section 3 assumes a lot of detailed knowledge, particularly about ML, on the part of the reader. Please work through it asking the question 'for each example of a trained network, have we explained what the input is and what the output is?

In presenting results, you should give some indication of what is a statistically significant difference. I know this is difficult in the case of other people's work.

I suggest you include graphs of relative performance gain against noise level.. do you get a bigger advantage for higher noise level?

Finally, you don't discuss how a system like your could be put into practical use? You can't expect speakers to have cameras focussed on their lips in normal circumstances, so are we talking about recordings to be processed later? Is this better than having a close-talking mike? If you have such a mike, will it obscure the lips?

Reviewer 3 Report

The paper is devoted to many aspects that are associated with deep machine learning technologies for solving problems that are associated with automatic processing and recognition of the speaker's audiovisual speech. It is clear from the authors' paper that today, automatic speech recognition systems are widely used in everyday life (for example, Siri, Cortana and others). However, the scope of such systems (relying solely on audio modality processing) is quite limited. Therefore, the authors of the paper also mention the fact that in acoustically noisy conditions (for example, on the street, at the train station, driving a car, in the subway, etc.), the speech recognition accuracy of such audio systems is rapidly deteriorating. Therefore, the authors of the paper propose to use cross-modal fusion through audiovisual analysis. For experiments, the authors chose two datasets (LRW and LRS2). The topic is undoubtedly relevant. On the basis of comparative analysis, it can be argued that qualitative indicators still depend on the input data, which in turn may be dependent on the selected data sets. It is also worth noting that the authors refer to A* level world conferences (ICASSP, INTERSPEECH, CVPR and others). However, there are questions and flaws.   1) First of all, it is striking that in the first part of the paper (section 2) the authors describe the already available methods for automatic speech recognition of the speaker. Examples of methods that have been trained and tested on available datasets (LRW, LRS2 and LRS3-TED) are provided. However, there are no references to the best works (methods and neural network models) on audio and video modality as such. This gap should be filled and a description should be added, for example 3-5 best results on the well-known LRW corpus (https://paperswithcode.com/sota/lipreading-on-lip-reading-in-the-wild). As long as it looks weird, in the description of related work there is a link to the work (link 20) and other, which is not even in the top 10 works, and the methods that have shown themselves the most are not considered. The same is recommended for the LRS2 dataset (https://paperswithcode.com/sota/lipreading-on-lrs2). 2) It should be explained. Why Dlib was used to localize the landmarks of the face, and not, for example, MediaPipe Face Mesh? After all, MediaPipe Face Mesh localizes a lot more facial landmarks (including the lip area). 3) There was also a question. LRW and LRS2 data were used for training, but LRW was not tested? If not done, why not? 4) Have you used any augmentation techniques (MixUp and others), cosine annealing? If not done, why not? More details needed. What doesn't fit and why? 5) For a better understanding of the drawings, it is recommended to submit them in vector formats (not png or jpg). 6) The style of the paper requires minor revision due to the presence of spelling and punctuation errors.   For the rest, it is worth noting that the paper is interesting, but only the points that are presented in the form of current shortcomings and issues are embarrassing. It seems to me that all the proposed additions will only improve this paper, and it will be useful and interesting to many specialists who associate their research with the automatic processing of audiovisual speech, but it is better to refine it first.

Round 2

Reviewer 3 Report

Flaw number 1 has not been fixed.

The authors added "For example, the proposed Visual-Audio Memory [16] that can recall the audio features with just using the input video, and it achieves the best performance on the LRW dataset so far, with an accuracy of 88.5%". This result is far from the best. It is currently only 4. The best 3 results so far are 94.1 (paper in arxiv), 89.52 (paper in ICASSP 2022), 88.7 (paper in EUSIPCO 2022).

See the same for the WER metric for the LRS2 dataset.

See ranking (https://paperswithcode.com/sota/lipreading-on-lip-reading-in-the-wild and https://paperswithcode.com/sota/lipreading-on-lrs2).

The article really deserves attention scientific community, but you need to write the current state of research.

Correct the flaw and the article can be recommended for publication.

Author Response

We feel great thanks for your professional review work on our article. According to your nice suggestions, we have made corrections to our previous draft.

Thank you very much for your careful inspection, we apologize for our carelessness, and we have revised the results of the current best lip reading model on the LRW and LRS2 datasets in the second part of the paper. And, references have been added and modified on related work on audio-visual speech recognition to make it more relevant to our research work.

Our previous work on related work mainly introduces the more representative methods in the current field, no introduction to our own methods, the main purpose of this section is a paragraph similar to the overview, introducing themselves to the frontier-related space written in the introduction section. We ignore whether the effect of the method is currently superior, and has been improved accordingly in the paper. And, you mentioned why you did not refer to the best lip reading model, based on the problem that one lip may correspond to more than one meaning in our presentation, which is also added in the second part of the paper.

Finally, you mentioned to elaborate our current research status. Combining this modification, although the performance of lip reading is far from the current best, there is a great improvement in the performance of audiovisual speech recognition, and our goal is to improve the performance of speech recognition in noisy environment, and the effectiveness of our model can be seen from the experimental results.